# Ion-Powered Rotary Motors: Where Did They Come from and Where They Are Going?

**DOI:** 10.3390/ijms241310601

**Published:** 2023-06-25

**Authors:** Vibhuti Nandel, Jacob Scadden, Matthew A. B. Baker

**Affiliations:** School of Biotechnology and Biomolecular Sciences (BABS), University of New South Wales, Sydney, NSW 2033, Australia; v.nandel@unsw.edu.au (V.N.); j.scadden@unsw.edu.au (J.S.)

**Keywords:** flagellar motor, motility, bacteriology, biophysics, ion-channel

## Abstract

Molecular motors are found in many living organisms. One such molecular machine, the ion-powered rotary motor (IRM), requires the movement of ions across a membrane against a concentration gradient to drive rotational movement. The bacterial flagellar motor (BFM) is an example of an IRM which relies on ion movement through the stator proteins to generate the rotation of the flagella. There are many ions which can be used by the BFM stators to power motility and different ions can be used by a single bacterium expressing multiple stator variants. The use of ancestral sequence reconstruction (ASR) and functional analysis of reconstructed stators shows promise for understanding how these proteins evolved and when the divergence in ion use may have occurred. In this review, we discuss extant BFM stators and the ions that power them as well as recent examples of the use of ASR to study ion-channel selectivity and how this might be applied to further study of the BFM stator complex.

## 1. Introduction

A molecular motor is a complex assembly of biomolecules like proteins, nucleic acids, or other molecules that perform mechanical work, such as movement, force generation, or information production [1,2,3]. Such motors are found in many biological systems and often operate through conformational changes in the motor structure that alter in response to a stimulus, such as a chemical or mechanical force [4,5].

Ion-powered rotary motors (IRMs) are examples of molecular machines that generate rotational motion in response to the movement of ions across the membrane. The three well-studied IRMs that exhibit high performance are the two ATP synthases, the F_O_/V_O_ motor in F-type ATP synthase and V/A-type ATP synthase, and the bacterial flagellar motor (BFM) [6,7,8,9,10,11]. The BFM utilises transmembrane stator proteins which are powered by ions to drive rotation and thus provide movement [12,13]. The most commonly studied stators are MotA-MotB and PomA-PomB, which are powered by protons (H^+^) and sodium (Na^+^) ions, respectively [14,15]. Preliminary evidence from *Aquifex aeolicus* has suggested that H^+^-driven motility had diverged from a Na^+^-powered ancestor [16], however further work is required in order to confirm this more generally. Other BFM stator units have been shown to use a variety of monovalent and divalent cations [17,18,19]. Recently it has been shown that bacterial complexes like ExbB-ExbD-TonB, GldLM, GldM/PorM, AglRQS and ZorAB use the ion motive force to exchange nutrients or power motility [20,21,22,23,24,25,26]. This variety seen in extant stator proteins and their functions, coupled with the variety of ions shown to be utilised to drive rotation, presents questions about how these proteins have adapted to various ions and how divergence occurred.

Study into the evolution of IRMs can show us how they have adapted over time in different bacterial species, which developed to occupy different niches. To investigate the evolution of ion selectivity in IRMs, particularly of the stator proteins of the BFM, an in-silico approach known as ancestral sequence reconstruction (ASR) could be used. This computational method relies on the analysis of extant amino acid sequences to predict the most likely sequences at particular nodes of a phylogenetic analysis [27,28]. This technique is becoming increasingly utilised not only in research but also in industry to develop new solutions for improved efficiency of molecular processes in protein engineering [29]. ASR can also be used to study the evolution of proteins and how and at what point in time divergence of function has occurred [30]. This technique may be particularly useful in the understanding of ion use divergence of BFM stator units.

In this review, we cover the structure and function of IRMs, particularly the BFM stator unit from *Escherichia coli* and *Vibrio* species, and then explore the role of ASR in elucidating significant evolutionary changes that may correlate with ion selectivity.

## 2. The Structure of the Bacterial Flagellar Motor

Motility is an important property that enables many bacteria to escape unfavourable environments or explore their surroundings in search of food. In polytrichous flagellated bacteria like *E. coli,* which has multiple flagella on its cell body, the flagella come together and bundle up [31]. The rotation of the flagellar bundle propels the forward or backward movement of *E. coli*. The disassembly of the flagellar bundle causes the bacteria to tumble around. Thus, the rotation of the flagella imparts a “run and tumble” motion to the bacteria [24].

Present at the base of the flagellum in *E. coli* is an approximately ~60 nm molecular motor, the BFM, which powers its rotation [32]. The BFM embeds in the cell envelope, which includes the cell membrane and cell wall. It is further anchored to the peptidoglycan layer of the cell wall by a series of protein complexes [6]. The supramolecular motor complex forms after ~22 proteins self-assemble in the cell wall [33]. Structurally, the flagellum is a propeller that helps bacteria manoeuvre through liquids and different surfaces in different directions. It is joined to the basal body by the hook, which acts as a joint between the two [34]. The basal body that spans the cell envelope is at the other end of the hook, constituting the rotor and stator units [6]. The core components of basal bodies demonstrate a conserved pattern, even though there is variation in the structures observed in different species across Gram-positive and Gram-negative bacteria (Figure 1). In *E. coli*, four rings encompass the rod: the LP ring (absent in Gram-positive bacteria [35]), the MS ring and the C-ring (Figure 1); an additional H-ring and T-ring are present in *Vibrio* [36,37,38].

The stator complex present at the inner membrane constitutes two proteins. Typically, these are MotAB for a H^+^-driven stator or PomAB/MotPS for a Na^+^-driven stator [12]. In addition, MotXY proteins are also present in *Vibrio*, and MotC, MotD, and MotE have also been reported in other bacteria [39,40]. Most bacteria possess a Mot-like H^+^-driven stator, as in *E. coli*, or a Pom-like Na^+^-driven stator, as in *Vibrio*, or both. A genomic data survey conducted shows that more than 60 bacteria possess two or more putative stator units. For example, *Bacillus subtilis* has H^+^-driven MotAB and a Na^+^-driven MotPS complex [41]. They utilise each stator depending on the environmental conditions.

The MotA subunit comprises four transmembrane (TM) domains, with the first two TM domains connected to TM3 and TM4 via a sizeable cytoplasmic domain, and TM4 ending with a short cytoplasmic C-terminal tail [12,13]. MotB, on the other hand, has a single N-terminal TM helix followed by the plug region that contributes to the active and inactive states of the stator units, and a large peptidoglycan domain in the periplasmic space [12,42]. Five units of MotA assemble symmetrically around two units of MotB (Figure 2), resulting in a 5:2 stoichiometry [12,13].

The interaction between some residues of stator subunits results in an ion channel formed by TM3 TM4 of MotA and the TM helix of MotB [12,13]. While these stators remain fixed into the bacterial cell wall, they can associate and dissociate from the rotor in response to external stimuli. During the resting state, the stator unit is usually inactive and disconnected from the motor. They become active upon motor incorporation and binding to the peptidoglycan layer [12,13,43,44]. The ion gradient across the membrane translocates the ions through the ion channel formed by the stator units. In *E. coli*, H^+^ ions pass through the ion channel, interacting with the conserved D32 residue of MotB, subsequently causing the rotor to rotate incrementally after a power stroke on the C ring [45,46,47,48]. Towards the end of this cycle, the H^+^ ions are released from the MotB subunit resulting in another conformational change [12,13].

### 2.1. Ion Selectivity

Among the bacteria with a bacterial flagellar motor, H^+^ or Na^+^ ions are commonly found to cross the ion channel formed by MotAB/PomAB. Some studies have argued that alternative ions can pass through and power the stators. In 2012, a study on *Bacillus alcalophilus* stated that potassium (K^+^) and rubidium (Rb^+^) could couple with the MotPS subunits of the BFM [17]. While in 2015, Imazawa et al., first reported divalent ions like calcium (Ca^2+^) and magnesium (Mg^2+^) were required to drive the *Paenibacillus* motor [18]. Later, another study by Onoe et al., showed that monovalent ions could be used to power the *Paenibacillus* stator units instead of divalent ions [19]. These studies highlight the diversity in the stator units. They also pose questions: While the stators are ion-specific, can they adapt to using different ions with the same efficiency as the H^+^ or Na^+^ ions? Are there more stators out in nature that use alternative ions, or were there at any time during evolution that utilized ions other than H^+^ or Na^+^?

Onoue et al., investigated the mechanism of ion selectivity for Na^+^ in the *Vibrio* stator PomAB. In addition to the conserved Aspartic acid residue on the N-terminal helix of the TMH of PomB, they identified two threonine (T158 and T186) residues that regulate the recognition of the Na^+^ ions [49]. In a recent preprint by the Taylor group, three threonine residues in Vibrio are reported to be quintessential for the conduction of the Na^+^ through the stator units (Figure 2) [50]. In addition to the above-mentioned amino acids, some other residues that have been speculated to play a role in the ion-conduction pathway in *Vibrio* are K64, F66 and M67 [51]. In 2020, apart from the role of the conserved D22 in MotAB, the H^+^ driven stator in the B subunit, Y20, F23 and S25, were identified to be essential residues in MotB for the selectivity of the H^+^ ions. In the A subunit, T189, F186 and T155 were observed to be essential residues [12] (Figure 2).

Experiments by Asai et al., ruled out MotA’s role in ion filtering, narrowing down the ion selectivity role to the B subunit [52,53]. Based on these studies, the researchers concluded that the periplasmic area of PomB, which is close to the inner membrane, potentially plays an essential role in ion specificity, perhaps by modifying the size of the TM pore [54]. To identify the specific regions in MotB that act as a filter, MomB, a chimera of the N-termini MotB and C-terminus of PomB from the *Rhodobacter sphareoidis* and *Vibrio alginolyticus*, was constructed and expressed in *V. alginolyticus*. The location of the MomB/PomB cut and the proportion of the B-unit that was MotB/PomB was observed to affect specificity [55]. Research on dual-ion conducting stator units shows that mutations near the surface of the conversed TM region of the B/S subunit play a role in: (1) ion preference, like H^+^/Na^+^ in *Bacillus clausii* or *Bacillus subtilis* [17,41]; (2) converting stator from multiple-ion coupling to single-ion coupling [17]. Evidence of ion selectivity was provided in the steered molecular dynamics study by Nishihara and Kitao, which used disulfide cross-linking and tryptophan scanning mutagenesis to build a MotAB model before structural models had been solved in 2020. This study suggested size exclusion of the ions as the ion selectivity mechanism [56].

Studies over the last two decades have reported the role of many residues in the stator complex which, when mutated, render the bacteria non-motile. Table 1 summarises some observed mutations with relevant effects on ion selectivity.

Recent research by Ridone et al., examined the adaptation of the stator units under different selective pressures. The study used as a model an edited *E. coli* where the native *motAB* genes corresponding to the H^+^-powered stator was replaced with *pomAB* genes [58] to generate a Na^+^ driven *Pots* strain. During the evolution of this strain in a K^+^ rich environment, mutations in the stator genes were observed and one mutant, PotB G20V, showed reversal to become H^+^-driven [59]. This work showed that selection for motile variants can be swift.

### 2.2. Towards New Power-Sources for IRMs

The bacterial world is not limited to H^+^ and Na^+^-rich environments. It is natural to ask if the bacterial diversity in, for example, salt-rich lakes has adapted to exclusively utilizing ions like Li^+^. Examples may include salt-encrusted basins known as salars, found in South America, the Black Sea, or the stratified lakes of Antarctica. In many of these environments, motility studies have not yet been executed and future studies of salt-rich lakes could help answer questions regarding how stator genes can evolve to sustain bacterial motility.

## 3. The Role of Ancestral Sequence Reconstruction in Understanding the Evolution of Ion Selective Motor Proteins

The process of evolution is fundamental to living organisms effectively exploiting the niches in which they reside. Predicting when protein functions diverged in evolutionary history can inform us how extant proteins have evolved and how functions have been gained or lost over time. The variety of ions used by extant stator proteins of the BFM prompts questions of why there is such variation in ion use and at what point in evolutionary history the divergence in ion use occurred.

### 3.1. Ancestral Sequence Reconstruction

Ancestral sequence reconstruction (ASR) is an in silico technique allows for ancestral proteins to be ‘resurrected’ and further used to assess their potential uses in academic, medical or industrial applications [27,60,61,62,63,64]. Modern extant homologue sequences are used to create an alignment after which the phylogenetic relationship is established through tree generation followed by statistical modelling, which provides predicted sequences at each node within the phylogenetic tree [28]. Sequences at each node of interest can then be selected for further investigation, whereby the predicted amino acid sequence can be used to re-create the nucleotide sequence that can then be cloned into an expression vector. Once cloned, the function of the protein can be assessed and compared in performance to extant proteins (Figure 3) [28].

ASR has many uses including homology detection, prediction of ancestral function, and investigation of the evolutionary divergence points of modern proteins [65,66]. It can be used to search for ancestral proteins with desirable characteristics such as increased thermostability or greater promiscuity [67,68]. However, ASR can also be used in order to investigate the evolution of proteins that have roles in intricate modern complexes, such as the bacterial flagellar stator protein MotB [69]. ASR can be used for ancestral protein investigations using both prokaryote and eukaryote systems (Table 2). A previous review of the literature has shown that ASR has been used for the study of a number of ancestral sequences of proteins, for example, ketol-acid, rubisco, β-lactamases, polar amino acid–binding proteins, cyclohexadienyl dehydratase and steroid receptors [27]. This indicates that ASR is a multi-disciplinary tool that can be used to study both multicellular and unicellular organisms.

### 3.2. The Use of ASR to Study Ion Selectivity in the BFM

ASR has been used to elucidate the evolution of ion selectivity of proteins. Research into the bacterial flagellar stator protein transmembrane B-subunit (MotB) used ASR to create 13 ancestral candidates that were then used in conjunction with the A-subunit (MotA) [69]. As discussed earlier in this manuscript, the B-subunit is crucial for ion selectivity and it was shown that all 13 B-subunit candidates were able to form functional stator complexes and restore motility to stator-deleted *E. coli* strains [69]. Flagellar rotation of ASR candidates was shown to be Na^+^-independent, and four of these ancestral stators were also able to form functional complexes with the A-subunits of *Aquifex aeolicus* and function in a Na^+^-independent manner [69]. The authors also showed that regardless of phenylalanine (F) or tyrosine (Y) at key residue position 30, proton-powered motility was observed [69]. This work helps to demonstrate that ASR can be used to identify ancestral candidate sequences that can be studied, and that function in combination with extant proteins in complexes. The converse experiment using ancestral MotA proteins in combination with extant MotB was also executed, and all ancestral MotAs were shown to require H^+^ for motility [83]. Of the 10 ancestral MotAs tested, four ancestral MotA were able to interact with extant MotBs and ancestral MotBs previously created by the authors [69]. The ionic power source for three of the functional proteins was tested and it was found that all the candidate ancestral MotAs required H^+^ and K^+^ for motility, and no swimming was observed if Na^+^ was used [83]. Additionally, the authors identified 30 critical residues across multiple domains of MotA. The authors concluded in this study that ASR could be used not only for analysis of the function of ancestral stator proteins, but they also have the potential to assess the evolution and conservation of critical residues within ancestral and extant stator proteins.

### 3.3. Limitations of ASR

The use of ASR to identify potential ancestral candidates must be treated with caution and an understanding that the sequences predicted from the phylogenetic analysis are only the best predictions made from available extant sequences. When assessing proteins for use in ASR pipelines, several factors need to be taken into account. The first is whether multi-domain proteins with only one conserved domain are being used; secondly, whether horizontal gene transfer has influenced the phylogeny; and lastly, the sample selection size of extant sequences [61,84]. Furthermore, due to the predictive modelling nature of ASR, the selection of residues might be biased towards proteins that function more efficiently than the authentic ancestral protein. Such bias may explain an increase in the thermal stability of expressed proteins via ASR [85,86]. The quality of alignment will also influence the outcome of the predicted ASR sequences. The use of incomplete and limited sequence data and ambiguity regarding gap placement can increase phylogenetic uncertainty and reduce the quality of the reconstruction [87].

ASR can be used to generate ancient protein candidates in conjunction with other techniques, such as experimental evolution. This could be particularly useful for questions surrounding the ion selectivity of the flagellar motor. It is still an outstanding question in evolutionary biology whether the ion selectivity in the BFM adapted to use specific ions over time or whether there was greater ion promiscuity in ancient flagellar stator complexes. By studying these ancestral BFM motor proteins as expressed proteins in model lab species, it is possible to execute rapid evolution assays to determine which changes in ion selectively can be observed rapidly.

## 4. Concluding Remarks

The BFM is required for bacterial motility. Studies have shown that the stator proteins of the BFM in many bacterial species are required for the generation of rotational movement, and that they use different ions. There are examples of ion-powered rotary motors in extant bacteria that are powered by multiple ion types. This poses the question: which ions were originally used for motility at the beginning of life and how has the divergence in ion use occurred since? ASR is a technique that can assist in answering these questions and future work will generate candidates that provide insight into how extant IRMs have evolved.

## Figures and Tables

**Figure 1 ijms-24-10601-f001:**
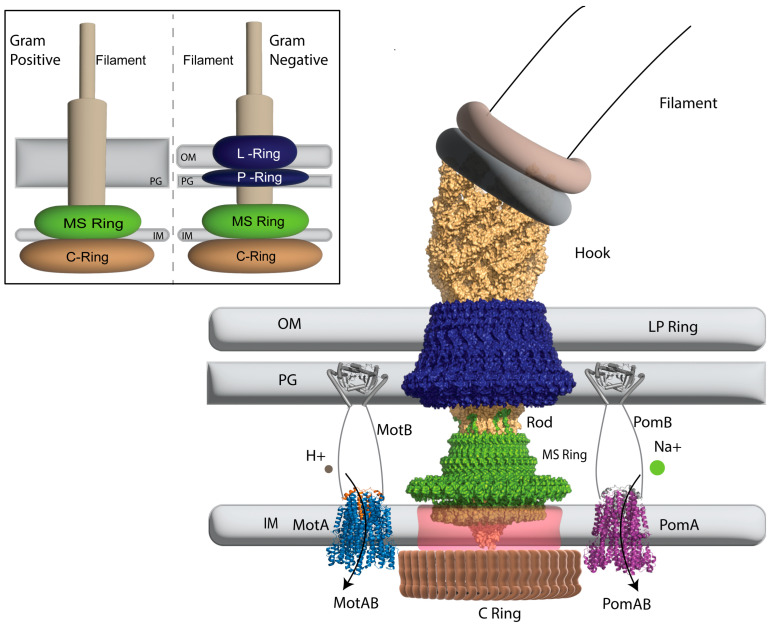
Semi-surface representation of the structure of the bacterial flagellar motor in a gram-negative bacterium with the LP ring in dark blue, MS ring in green, and stators represented as cartoon cylinders: MotA_5_B_2_ (PDBID: 6YKM) in blue-orange with five MotA (blue) subunits assembled around the MotB (orange) dimer on the left, and a representative example PomA_5_B_2_, in purple, on the right. The peptidoglycan binding domain of MotB/PomB (light grey) extends from the transmembrane units to the peptidoglycan domain of MotB/PomB. The Hook and Rod are represented in light brown, with the C Ring in dark brown. Arrows represent direction of ion travel. OM: Outer Membrane; PG: Peptidoglycan Layer; IM: Inner Membrane. Basal body/hook PDBID: 7CGO; stator: PDBID: 6YKM. Inset is an illustrative comparison of the bacterial flagellar motor in Gram-positive bacteria (**left**) and Gram-negative bacteria (**right**), highlighting the differences in the cell-wall composition and the Rings in the BFM.

**Figure 2 ijms-24-10601-f002:**
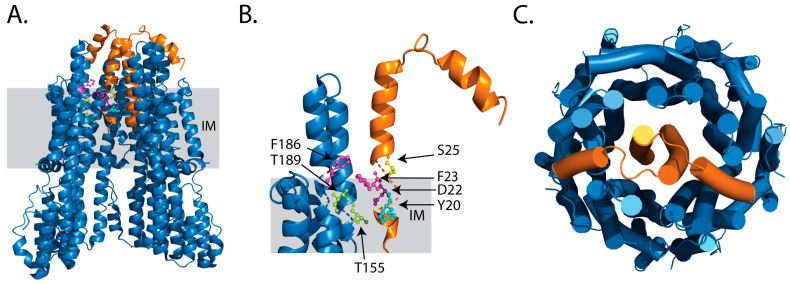
Structure of MotAB from *Campylobacter jejuni* (PDB: 6YKM). (**A**) The longitudinal view showing four out of five MotAs in blue and the MotB dimer in orange with important residues involved in ion-selectivity as ball and stick. (**B**) Close-up of the Monomer of the MotA (blue) and MotB (orange) subunit with the conserved D22 in MotB in purple, threonine (T) in green, phenylalanine (F) in magenta, tyrosine (Y) in cyan and serine (S) in yellow, representing the residues important in the transport ion pathway. (**C**) the top-view representing five chains of MotA and the dimer of MotB, respectively. IM: Inner Membrane.

**Figure 3 ijms-24-10601-f003:**
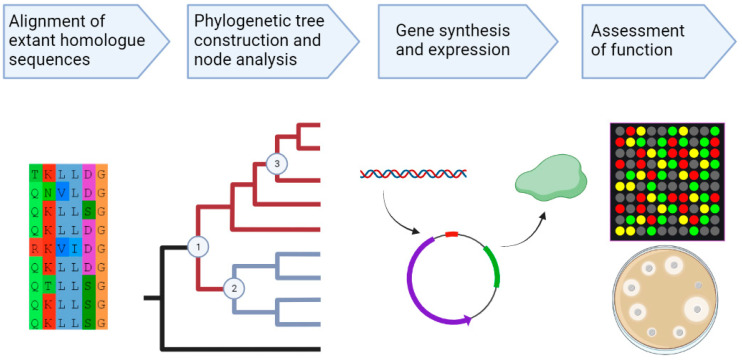
A flow-diagram illustrating the process through which ancestral sequence reconstruction is performed. This is followed by gene synthesis of the nodes of interest which allows for assessment of the functionality of the protein once it has been expressed and comparisons to extant protein descendants.

**Table 1 ijms-24-10601-t001:** Some examples of mutations that affect the ion selectivity and usage.

Organism	Stator	Mutation	Ion Selectivity	Effect	Reference
*Bacillus alcalophilus*	MotPS	MotS-M33L	Na^+^/K^+^/Rb^+^	Loss of K^+^/Rb^+^ coupling motility-*E. coli*	[17]
*Bacillus clausii*	MotAB	MotB-V37L, A40S, G42S	Na^+^	Only Na^+^ selective	[41]
		MotB-G42S, Q43S, Q46A	H^+^	Only H^+^ selective	
*Vibrio algynolyticus*	PomAB	PomA-S25C	Na^+^	Reduce motility	[53]
		PomA-D31N		Affect ion usage	[57]

**Table 2 ijms-24-10601-t002:** Recent examples of proteins that have undergone ancestral sequence reconstruction and had their ancestral sequences tested for function. The versions of phylogenetic and ancestral sequence reconstruction software used in each study are provided if stated in the cited literature.

Protein	Number of Aligned Sequences	Phylogenetic/Ancestral Reconstruction Software Used	Key Findings	Reference
MotB	757	Quicktree/PAML	Ancestral MotB proteins were Na^+^ independent and could interact with extant MotAs	[69]
SARS-CoV-2	14,164	IQ-TREE (v2.1.2)/TreeTime (v0.9.5)	Analysis of mutants through ASR identified that strain BW.1 arose in the Yucatan Peninsula	[70]
Flavin-dependent monooxygenase	276	PhyML (v3.0)/PAML (v4.8)	Two mechanisms were identified to control stereochemical outcomes of the oxidative dearomatization reaction	[71]
Orange carotenoid protein (OCP)	189	PhyML (v3.1)/PAML (v4.9)	Showed that the ancestral OCP regulator was horizontally acquired by cyanobacteria then co-evolved and that pre-HGT protein could still interact	[72]
Xylose isomerases (XI)	1042	Mega11/PAMLX	Amino-terminal fragment of ancestral XIs were important for maintaining activity and high performing enzymes were found that could contribute to high ethanol titers	[73]
Activation-induced cytidine deaminase (AID)	71	RAxML (v8.2.9), MrBayes (v3.2.7)/ProtASR (v2.0 and 2.2)	Enzymatic inactivation of reconstructed AIDs took place recently in the Atlantic cod family and thus explains that lack of secondary immunity in cod	[74]
Influenza A	3443	RAxML (v8.0), Geneious (R9.0.3)/Lazarus (v2.0)	Analysis of reconstructed swine influenza viruses from 1979–1992 showed that transmission in piglets was enabled by changes in viral polymerase protein and nucleoprotein since 1983	[75]
Cytochrome P450 family 1 enzymes (CYP1s)	471	Mega11/GRASP	Younger ancestors were shown to have activities toward xenobiotic and steroid substrates than older ancestors. Greater thermostability was seen in older ancestor CYP1s, however caffeine metabolism was shown to be a recently evolved trait	[76]
PETase	914	IQ-TREE2/GRASP and PAML	Two ASR candidates were shown to have higher catalytic activity and thermostability was also increased	[77]
Coagulation factor IX (FIX)	59	MrBAYES/PAML (v4.1)	Ancestral FIX variants were shown to have enhanced activity and that AAV-ancestral FIX Padua vectors had greater potency over AAV-human FIX Padua vectors in haemophilia B mice	[78]
Phytohormones-CYP711A	346	RAxML/GRASP	Reconstructed CYP11As accepted GR24 as a substrate and the monocot group 3 ancestor showed increased catalytic activity and high stereoselectivity towards GR24	[30]
Prion Protein (PrP)	161	PhyML (v3.3.2)/PAML (v4)	Aggregation of the PrP from the oldest ancestor was observed however ancestral bird PrP could not be seeded with extant prions. Ancestral primate PrP could be converted with all prion seeds	[79]
Fe/Mn superoxide dismutases (SODs)	738	IQ-Tree/PAML and FastML	Fe/Mn SODs were shown to be able to bind to both Fe and Mn, whereas extant SODs have been shown to have specific affinity to one ion	[80]
Nicotinic acetylcholine receptor (α9α10)	52	Mega5/PAML	Three residues were found in the α9 subunit which increased Ca^2+^ permeability for mammalian receptors but not for avian receptors	[81]
Family I.3 lipase	83	RAxML/PhyML	There was a deletion of residues during the evolution of this protein. Mg^2+^, Rb^+^ and Zinc (Zn^2+^) ions were also able to increase the relative activity indicating greater promiscuity of the ancient protein	[82]

## Data Availability

No supporting data.

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
