# Peer review of "Ion-Powered Rotary Motors: Where Did They Come from and Where They Are Going?"

_ijms, 2023, doi:10.3390/ijms241310601_

Round 1

Reviewer 1 Report

Review by Nandel et al. about ion-powered rotary motors that specifically focusses on the bacterial flagellar motor is well-written and a valuable addition to the field. Authors describe the examples of ion-powered rotary motors in biology and then explain the bacterial flagellar motor in detail. The review does a good job of giving a good overview of the different ion-powered bacterial flagellar motors. Finally, the application of ancestral sequence reconstruction to ion selectivity in bacterial flagellar motors is discussed. Following are my major comments addressing which will help in improving the manuscript.

1.      Figure 2 needs to be remade, especially panel A by zooming much more closely on the residues that are thought to be involved in the ion selectivity with the amino acids represented as ball and sticks. Color choices can also be revised so that MotA and MotB are distinct.

2.      The review on how ancestral sequence reconstruction has been used to study other systems is very valuable and the table format is easier for the reader. However, since the review focuses on the ion selectivity of the flagellar motor the authors should elaborate the work by Islam from 2020 and 2023 in more detail. Lines 233 -256 can actually be added to the table as discussing other systems in detail is not the focus of the paper.

Author Response

Figure 2 needs to be remade, especially panel A by zooming much more closely on the residues that are thought to be involved in the ion selectivity with the amino acids represented as ball and sticks. Color choices can also be revised so that MotA and MotB are distinct.

We thank the reviewer for their time reading our work and their  suggestions for improvement.

We have now remade Figure 2 (Line 159). We have altered the colours used for MotA and MotB to match those used in Figure 1 to make the overall colour scheme of the paper more coherent. We have improved clarity of the specific residues using ball and stick representations, as requested, and shown this as a zoom of the MotA-MotB interactions at the pore. We agree this is a good idea and improves understanding of these critical residues and feel it is now presented more appropriately. Thank you for the suggestion for improvement.

 The review on how ancestral sequence reconstruction has been used to study other systems is very valuable and the table format is easier for the reader. However, since the review focuses on the ion selectivity of the flagellar motor the authors should elaborate the work by Islam from 2020 and 2023 in more detail. Lines 233 -256 can actually be added to the table as discussing other systems in detail is not the focus of the paper.

We have now added lines 233-256 to Table 2 and discussed in further detail the work performed by Islam (2020 and 2023) (Lines 224-247). As this was our work, we were originally cautious to spend too much time reviewing our own work for fear of appearing biased, but we agree we probably leaned too far the other way and it is appropriate given a flagellar motor focus to give more detail here which we have now done. The paper now describes in more detail the use of ASR in studying the bacterial flagellar motor.

Reviewer 2 Report

The manuscript is scientifically sound. It has potentially high interest to readers interested in the area of bacterial flagellar research. I would suggest that the authors consider the following points as they revise their manuscript.

The introduction is poorly referenced with literature that does not support the statements made in the text.

The introduction needs minor revision. They could check for recently published articles; please add a specific introduction that will perfectly match. 

Figure 1: kindly add the gram positive bacterial flagella also.

Minor editing of English language required

Author Response

The introduction is poorly referenced with literature that does not support the statements made in the text.

The introduction needs minor revision. They could check for recently published articles; please add a specific introduction that will perfectly match. 

Figure 1: kindly add the gram positive bacterial flagella also.

We thank the reviewer for this suggestion and we have added multiple references to the introduction. We have also altered the introduction so that it relates more to the subsequent text in the review. The introduction is more specific to ion rotary motors and how we can use ancestral sequence reconstruction as a tool for studying the evolution of the motors, in particular the bacterial flagellar motor.

We have now added an inset to Fig. 1 (Line 80) which shows a comparison in structure between the Gram-positive and Gram-negative flagella. This primarily highlights the lack of L- and P- ring in the gram positive species.

We thank the reviewers again for their time to review the manuscript.